# Assessment of the Most Impactful Combination of Factors Associated with Nocturia and to Define Nocturnal Polyuria by Multivariate Modelling

**DOI:** 10.3390/jcm9072262

**Published:** 2020-07-16

**Authors:** Tine Olesen, Jerome Paul, Pierre Gramme, Marcus J. Drake, Johan Vandewalle, Karel Everaert

**Affiliations:** 1Urology Department, Ghent University Hospital, 9000 Ghent, Belgium; Karel.Everaert@uzgent.be; 2DNAlytics, 1348 Louvain-la-Neuve, Belgium; jerome.paul@dnalytics.com (J.P.); pgramme@gmail.com (P.G.); 3Bristol Urological Institute, University of Bristol, Bristol BS105NB, UK; marcus.drake@nbt.nhs.uk; 4Department of Pediatric Nephrology, Safepedrug, University Hospital Ghent, 9000 Ghent, Belgium; johan.vandewalle@uzgent.be

**Keywords:** nocturia, nocturnal polyuria definition, multivariate modelling, combination of factors impacting nocturia

## Abstract

Background: Nocturia is common and associated with multiple disease states. Many potential mechanisms have been proposed for nocturia, which also remains challenging to manage. Purpose: To use multivariate analysis to determine which combinations of factors can accurately discriminate clinically significant nocturia in patients to facilitate clinical management and treatment decisions. Patients and methods: Data analysis was based on frequency volume charts from three randomized controlled trials. There were 1479 patients included, of which 215 patients had no/mild nocturia and 1264 had clinically significant nocturia with at least two voids per night. Factors studied that may influence nocturia were demographics, sleep duration, functional bladder capacity, 24 h urine volume and literature-suggested definitions of nocturnal polyuria. We used univariate analysis and cross-validated multivariate modelling to assess association between factors and nocturia status, redundancy between factors and whether the combined use of factors could explain patients′ nocturia status. Results: The multivariate analyses showed that the most useful definitions of nocturia are ’Nocturia Index’ (NI) and ‘Nocturnal Urine Production per hour’ (NUPh) in combination with functional bladder capacity and sleep duration. Published definitions providing binary nocturnal polyuria outcomes had lower performance than continuous indices. These analyses also showed that NI was not specific to nocturnal polyuria as it also captured nocturia due to low functional bladder capacity. By contrast, NUPh was demonstrated to be specific to nocturnal polyuria. Conclusion: NUPh has previously been shown among elderly males to be essential in nocturia and a very valid measure of nocturnal polyuria. However, the current, large and independent dataset now confirms that it can be applied in an adult population with a complaint of nocturia covering both males and females.

## 1. Introduction

Nocturia, or waking up at night to void, is common. Two or more voids per night lead to impaired sleep health and lower health-related quality of life [1]. Thereby, the occurrence of two or more voids per night represents the threshold above which nocturia shifts from being perceived as a minor inconvenience to a bothersome and clinically meaningful condition, also resulting in an economic burden for society [2]. Nocturia is associated with multiple disease states [3]. Factors underlying nocturia may differ between individual patients but may also coexist within one patient.

In essence, the nocturia symptom is a result of the imbalance between functional bladder storage capacity (FBC) and urine production. Impaired FBC leads to small voided volumes and can be caused by underlying conditions, such as idiopathic/neurogenic detrusor overactivity and bladder outlet obstruction due to benign prostate enlargement (BPE). Treatments to improve FBC include beta-3 adrenergic agonists or anticholinergics in patients with detrusor overactivity and overactive bladder (OAB) symptoms, and α-blockers or 5α-reductase inhibitors in cases of BPE. The evidence for the effectiveness of these treatments in nocturia is limited [4,5,6,7].

Increased urinary output can occur overnight only or during a 24 h period. It can develop for many different reasons including lifestyle factors, heart failure, diabetes insipidus or mellitus, decreased vasopressin levels, venous insufficiency, or obstructive sleep apnea [8]. Interventions directed to reduce nocturnal urinary output include evening fluid restriction, leg elevation during daytime, timed diuretics, and antidiuretic treatment with desmopressin [9,10]. In fact, one study has shown that nocturnal polyuria (NP) was the most common contributor to nocturia [11].

There are different definitions of NP, all with limited evidence and varying specificity and sensitivity. The International Continence Society (ICS) issued, in 2002, a definition of NP [12] and the recent terminology update in 2019 also mentions the NP definition from ICS [13]. This definition—as well as other definitions—were later criticized due to their lack of accuracy in identifying nightly excess of urine resulting in the symptom of clinically significant nocturia [14].

This suggests multiple causes of nocturia in co-existence, and therefore, the pathophysiology of nocturia should not be reduced to one of a handful of basic mechanisms. Because of the multifactorial etiology, we hypothesized that statistical multivariate modeling [15] may be relevant as a tool to explain clinically significant nocturia. We aimed to determine which factors were accurately associated with nocturia. We also tested the ability of proposed definitions of NP, both as continuous and binary variables, to explain nocturia. Finally, we evaluated NP definitions as associated with nightly excess of urine, with the overall aim of facilitating clinical management and optimizing individual treatment decisions.

## 2. Materials and Methods

### 2.1. Study Design and Protocol

This is a post-hoc analysis based on screening data from three prospective phase 3, randomized controlled trials (000085/NCT01729819, CS40/NCT01223937, CS41/NCT01262456, see ClinicalTrials.gov) conducted to test the efficacy/safety of desmopressin in 2575 patients in Europe and North America (Table 1). As these analyses were based on screening data, patients had received no trial interventions prior to data collection. The main inclusion criteria for the screening phase was a complaint of nocturia in patients above 18 years of age. Data collection was based on 24 h frequency volume charts (FVC), collected for 72 h.

For the current study, we excluded patients (*n* = 391) with comorbidities outside the functional urology therapy area which may contribute to nocturia but which should be addressed by specialists in the relevant field before treating nocturia specifically [16]. This included conditions such as uncontrolled hypertension (*n* = 49), uncontrolled diabetes mellitus (*n* = 37), severe urgency and incontinence (*n* = 82), history of urological malignancies during the past 5 years (*n* = 9), polydipsia (*n* = 71), cardiac failure (*n* = 3), sleep apnea (*n* = 24), hepatic and biliary diseases (*n* = 43), current bladder stone or infection (*n* = 17), neurogenic detrusor overactivity (*n* = 2), alcohol and substance abuse (*n* = 4), renal impairment (*n* = 14). We also excluded those who withdrew consent (*n* = 37) and those with missing FVC (*n* = 670). Finally, those of American Indian and Asian of Hawaii or Pacific Island ethnicity (*n* = 35) were excluded as there were no matched controls in the no/mild nocturia) patient group for comparison. Our final analysis included 1479 patients of whom 215 had no/mild nocturia (less than 2 voids) and 1264 had clinically significant nocturia (at least 2 voids per night).

Factors proposed to influence nocturia were divided into three categories:
(1)Demographics, defined as:
-Age-Sex-RaceBody mass index (BMI)(2)Factors pertaining to fluid intake and urine output in relation to volume and time. These were denominated intake/output factors, and included:
-Functional bladder capacity (estimated based on ‘maximum voided volume’)-24 h urine volume-Sleep duration-Liquid intake within 1 h before going to bed(3)NP definitions already proposed in the literature. These were tested as either continuous or binary threshold variables (in total, 12 definitions):
-NUPh (nocturnal urine production per hour): ratio of nocturnal voided volume * to night duration-NUPh > 54 mL/h, NUPh > 78 mL/h and NUPh > 90 mL/h [17]-NUPw (nocturnal urine production per weight): ratio of nocturnal voided volume * to body weight-NUPw > 10 mL/kg [18]-NPI (nocturnal polyuria index): ratio of nocturnal voided volume to 24 h total voided volume-Age > 65 and NPI > 0.33 or Age > 20 and NPI > 0.2 [19]-NI (nocturia index): ratio of nocturnal voided volume to functional bladder capacity (i.e., max voided volume)-NI > 1.5 or NI > 1 [20]-NPIh: ratio of NUPh by DUPh (daytime urine production per h per 24 h)* Including first morning void.

These definitions were originally developed using a diverse set of outcome measures and based on study cohorts with limited sample sizes representing a subset of the general population. The definitions were found to be inaccurate with either high sensitivity but low specificity or vice versa when they were evaluated in a different or larger population. There were several papers that suggested major covariates exerted an important influence on the precision of the definitions. Currently, there is no optimal definition of nocturnal polyuria that offers accurate diagnostic value when tested in a general population [21,22,23,24,25,26,27,28,29,30,31].

Table 2 below shows the patients’ baseline characteristics. Patients had a median age of 59 years (interquartile range (IQR): 48–68) with 65% being females. The median number of nocturnal voids was for the no/ mild nocturia patients 1.5 (IQR: 1.0–1.7) and for the clinically significant nocturia 2.7 (IQR: 2.2–3.3). The NP definitions were applied to the population. In the table, “True” means congruence with the definition of NP and “False” means no congruence with the NP definition. For comparison, in the group of 670 patients that were excluded due to no FVC data, the median age was 66 years (IQR: 53–72) and there was a higher proportion of males (54.78%), but ethnicity and BMI were similar to the group included in the study (24.78% were African American; BMI: 28.60 (25.45–33.15)).

### 2.2. Statistical Analysis

‘R’ software was used for all analyses (https://www.r-project.org/).

Descriptive statistical analysis was used to assess whether different variables could discriminate clinically significant nocturia from no/mild nocturia. Non-parametric Wilcoxon rank-sum tests were used for continuous variables and Fisher′s exact tests were applied on categorical variables. *p*-values were corrected for multiple testing using Benjamini–Hochberg procedure [32].

We used cross-validated predictive model learning to assess whether multivariate combinations of the different factors could accurately discriminate clinically significant nocturia from no/mild nocturia [14]. Predictive models were repeatedly trained from a random sub-sample of the dataset and performance was assessed on a distinctly separate data set from the remaining sample (to evaluate its generalization capability). The discrimination was reported as mean Area Under the Receiver Operating Characteristic Curve (AUC) with 95% confidence interval [32]. Candidate predictive models were: Random Forest (RF, a non-linear, potentially complex model) and Regularized Logistic Regression (LR, a simple and robust linear model).

A total of 26 candidate signatures (i.e., lists of predictor variables) were evaluated:
Direct factors:-12 signatures, each including one NP definition and all four intake/output factors-One signature with the four intake/output factors, excluding NP definitionsAll factors (direct and indirect):-12 signatures, each including one NP definition, all four intake/output factors, and all four demographic factors-One signature with the four intake/output factors and four demographic factors, excluding NP definitions

Finally, we determined the correlation between NP and the predicted probability of clinically significant nocturia, and analyzed the models′ weights to determine the role of NP in the best linear associated model.

### 2.3. Ethics Approval and Consent to Participate

All participants gave written informed consent at the time of recruitment, and the project was approved by the local IRB/Ethics Committee.

## 3. Results

In univariate analysis, FBC, sleep duration, all proposed definitions of NP using both continuous and binary variables, gender at birth and race discriminated no/mild nocturia from clinically significant nocturia patients. Fewer variables tended to be identified as significant in analyses based on a per study basis. Details are provided in Appendix A.

Nocturia models based on intake/output factors and NP definitions discriminated clinically significant nocturia from no/mild nocturia well, as shown in Figure 1.

The model based on NI and additional intake/output factors had highest discrimination capability, followed by models based on NUP-indices, finally followed by NPI-indices. Our analysis also showed that models using NI or NUPh as a continuous NP definition combined with intake/output factors discriminate well (AUC: 82–84%, 77–80%, respectively). All models using binary definitions of NP had lower accuracy compared to those based on the corresponding continuous indices. Still, NI > 1.5 had high discriminatory accuracy (AUC:77–80%). Notably, the nocturia model using NPI as a binary definition combined with intake/output factors had lower association power than any other model including models that did not include any definition of NP. In general, nocturia models that added the demographic factors did not further improve the accuracy as other factors mediated these associations.

The association power varied between studies. Although the study cohorts of CS40 and CS41 were based on identical protocols, they differed in that CS40 was performed in women and CS41 in men; CS40 provided higher discrimination capabilities.

The continuous definitions NI and NUPh discriminated the patients with clinically significant nocturia from those with no/mild nocturia best. To investigate further which definition was most strongly associated with excess nightly urine production, we calculated the predicted probability of nocturia as a function of NI or NUPh, based on multivariate logistic regression (Figure 2).

The figure for NI as a function of nocturia was perfectly shaped to identify nocturia, with a correlation of 96%, which explained why no other intake/output factors were required to identify nocturia. Using a tentative NI cut-off of 1.5 showed a 50% probability of clinically significant nocturia, and NI values above 1.5 increased the probability of having clinically significant nocturia. Therefore, the NI parameter identified the risk of nocturia as a bothersome symptom but was not specific to an excess of nightly urine production. Applying the same principle to NUPh showed that the correlation was only 37% when using NUPh to identify nocturia. However, when NUPh was combined with bladder capacity and sleep duration, the AUC was 80%, i.e., similar to the AUC for NI alone.

## 4. Discussion

The management of nocturia patients is challenging. The etiology is of mixed origin and there is little available evidence about the effects of possible combinations of treatment modalities. However, the reported multivariate analyses provide insights into each factor’s capability to discriminate patients with clinically significant nocturia from those with no/mild nocturia. It, therefore, helps the treating physician to characterize each etiological factor, thereby facilitating treatment choices that can maximize the benefit for patients.

The recognized definitions of NP used today are binary, i.e., a fixed threshold for having NP or not. Our results strongly suggest that the binary approach to diagnose NP is suboptimal. This is consistent with the recognition that nocturia is a multifactorial symptom where each factor contributes with a different magnitude of bother. The currently used binary definitions of NP all had lower discrimination power than the continuous versions of the definitions to identify clinically significant nocturia. Notably, NPI, which is the most widely used definition, showed the lowest discriminative ability for clinically significant nocturia.

Whereas the most discriminating models for clinically significant nocturia used NI as the definition of NP, it is debatable whether NI is a definition of NP, or rather of nocturia. Based on the NI formula [3] (i.e., the ratio between nightly voided volume and FBC), FBC is the factor that determines NP-risk. Therefore, it may indicate a high risk of NP due to a small FBC. NI is, therefore, likely not to be specific to NP, which means it is not appropriate as a definition of excess urine production at night (NP). Instead, the NUPh parameter might be more appropriate and specific to NP. Being able to differentiate those nocturia patients for whom NP is a contributing factor is important in understanding how to manage the condition and reduce the bothersome symptom. Fulfilling a diagnostic criterion of NP would require treatment with, e.g., an antidiuretic, whereas low FBC would more likely benefit from treatment with an anticholinergic or a beta-3 adrenergic agonist. We, therefore, suggest use of NUPh as specific to NP. This measure has previously been shown to be valid in elderly males, but it has now been shown to apply to the full age range of adult males and females in this independent dataset. Since the aim of this analysis was to find a discriminative model for clinically significant nocturia, we have determined that NUPh provides a reliable model when combined with FBC and sleep duration as separate factors.

In the multivariate model, including demographic factors did not increase the predictability. This simplified the model and the information needed, as input for this model is also easily accessible for the clinician. The NUPh parameter also does not require information on daytime urinary volumes. However, it is still clinically important to collect this information using the FVC so that 24 h polyuria, not confined to the night time, can be identified if present, and dealt with appropriately before considering treatment strategies for nocturia and nocturnal polyuria specifically. The NUPh model is, therefore, a helpful diagnostic criterion in clinical practice. Our study was a post hoc analysis based on individual data in a pooled analysis. Although the sample size was quite large, this may constitute a bias in the study populations analyzed. However, there were similar trends in the individual studies as in the pooled analysis. Despite some minor differences between females and males, the model was applicable to both.

The population in this study was included based on a complaint of nocturia, i.e., the primary selection criterion was related to feeling bothered by nocturnal voids, independent of the number of voids. As a result, very few patients in the dataset had no nocturnal voids, even in the comparison group with no/mild nocturia. It can be speculated that the contrast in nocturnal urine volume between those with clinically significant nocturia and no/mild nocturia would become more explicit if more patients with no nocturia were added to the comparison group. However, the current dataset is a good representation of the clinical reality in that patients who seek help for nocturia experience the symptom to some degree and it is the physician’s task to differentiate those with clinically significant symptoms and the etiology of their condition.

It is also important to note that we excluded patients with chronic diseases associated with nocturia, such as uncontrolled diabetes mellitus, uncontrolled hypertension, previous malignancies and sleep apnea. The model is thus designed to apply to the specific subpopulation of nocturia patients who have functional urological diseases. The model should, therefore, be utilized once other underlying conditions and comorbidities—not relating to functional urology—have been excluded or treated accordingly [16].

## 5. Conclusions

To our knowledge, this is the first time all published definitions of NP have been tested in a large dataset of adult men and women with a complaint of nocturia.

All NP definitions (continuous and binary) showed significant associations with clinically significant nocturia and no/mild nocturia, as demonstrated in the univariate analysis. Sleep duration, bladder capacity, gender at birth and race also showed a significant association. However, multivariate analyses showed that the association of gender and race to nocturia status was mediated by the intake/output factors.

The multivariate analyses showed that the most discriminative definitions of NP were NI and NUPh. The previously proposed binary NP definitions performed more poorly than the equivalent continuous indices. However, these analyses also highlighted that NI was not specific to excess of nightly urine production, as it also captured nocturia due to small FBC. In contrast, NUPh seemed to be a definition of excess of nightly urine production alone, leading to good performance when combined with bladder size and sleep duration as a marker for clinically significant nocturia. This confirms a previously published finding in elderly males using the NUPh parameter, and extends its applicability to females and a younger population.

Therefore, we recommend using continuous NUPh (in the absence of 24 h polyuria), without applying a threshold for disease, as specific to an excess of nightly urine production for a general population that has a complaint of nocturia. This finding is important for the understanding of each treatable contributing factor and, therefore, for the selection of effective treatment for nocturia.

## Figures and Tables

**Figure 1 jcm-09-02262-f001:**
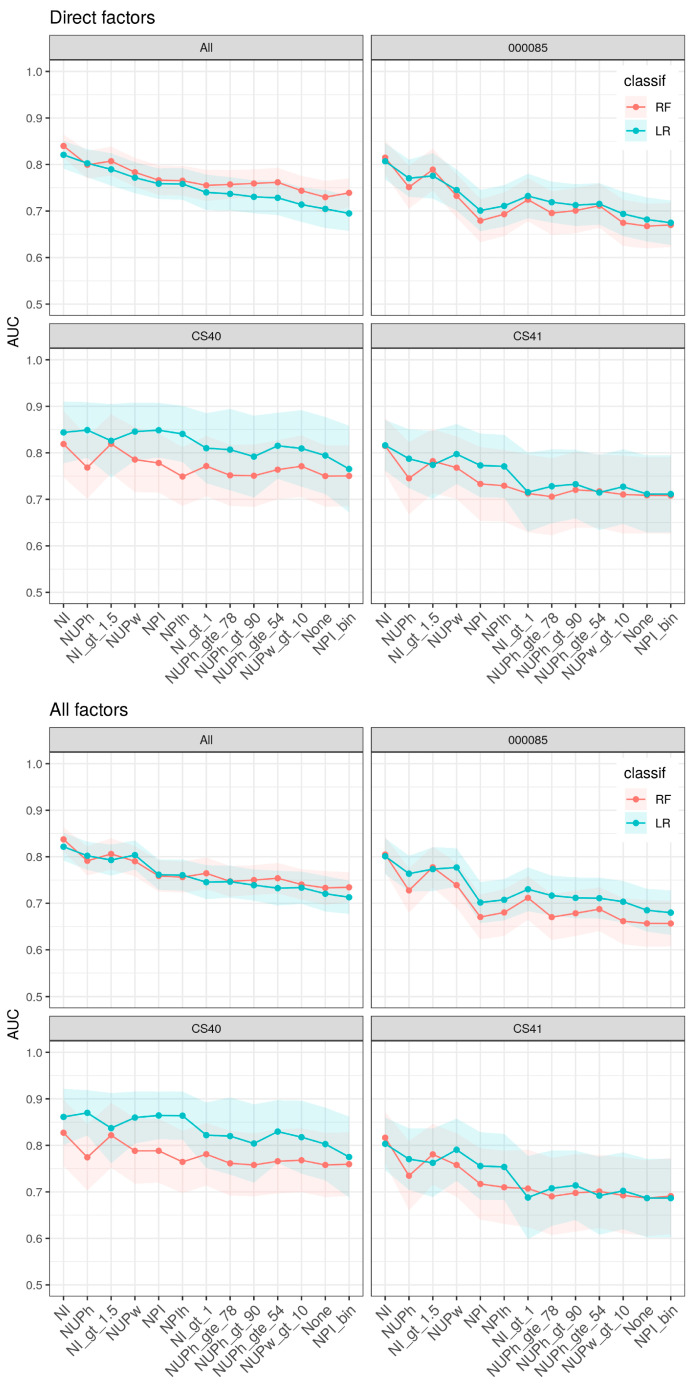
Pooled and per study analysis of nocturia versus non-nocturia discrimination factors. Each curve reports the AUC (*y*-axis) for different signatures (*x*-axis) and two different classifiers (two colors). AUC: Area under the curve; bin: binary; classif: classification; gt: greater than; gte: greater than or equal to; LR: logistic regression; NI: nocturia index; NUPh: nocturnal urine production per h; NUPw: nocturnal urine production per weight; NPI: nocturnal polyuria index; RF: random forest.

**Figure 2 jcm-09-02262-f002:**
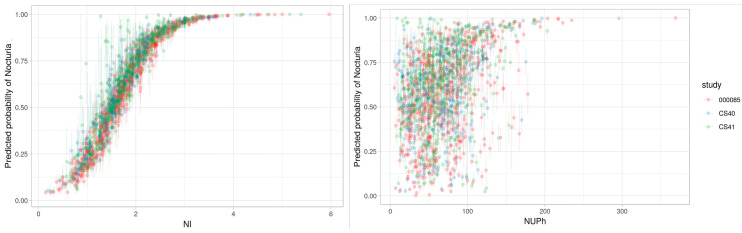
Correlation between nocturia and NI (**left**) and NUPh (**right**). NI: nocturia index; NUPh: nocturnal urine production per h.

**Table 1 jcm-09-02262-t001:** Patients included in the studies.

Excluded from the Analysis/Trial:	000085	CS40	CS41	Total
Total no. of patients	913	649	1013	2575
Exclusion criteria	200	81	110	391
Race	8	7	20	35
Missing Frequency Volume Chart	114	189	367	670
Remaining in the analysis	591	372	516	1479

**Table 2 jcm-09-02262-t002:** Baseline characteristics.

Variable		Summary for No/Mild Nocturia Patients Median (Q1–Q3)	N	Summary for Clinically SignificantNocturia Patients Median (Q1–Q3)	N
AGE (*y*)		56.00 (47.50–66.50)	215	60 (48–68)	1264
Gender at Birth	Female	(76.28%)	164	(63.21%)	799
	Male	(23.72%)	51	(36.79%)	465
Race	African American	(30.7%)	66	(23.66%)	299
	White	(69.3%)	149	(76.34%)	965
BMI		29.00 (25.40–33.55)	215	29.20 (26.00–34.10)	1264
Nocturnal Void (#)		1.50 (1.00–1.67)	215	2.67 (2.33–3.33)	1264
Max Voided Volume/Functional Bladder Capacity (mL)		355.00 (281.25–500.00)	214	300 (200–420)	1259
Volume 24 h (mL)		1300.00 (920.62–1771.58)	186	1390.00 (966.67–1875.83)	1176
NUPh (mL/h)		53.50 (37.19–77.97)	206	65.18 (43.50–90.96)	1226
NUPh > 90	TRUE	(31 (15.0%))		(314 (25.6%)) (765 (62.4%))	
	FALSE	(175 (85.0%))		(912 (74.4%))	
NUPh > 78	TRUE	(52 (25.2%))		(449 (36.6%))	
	FALSE	(154 (74.8%))		(777 (63.4%))	
NUPh > 54	TRUE	(102 (49.5%))		(765 (62.4%))	
	FALSE	(175 (85.0%))		(461 (37.6%)) (912 (74.4%))	
NUPw (mL/kg)		5.40 (3.45–7.59)	214	6.82 (4.47–9.65)	1241
NUPw > 10	TRUE	(26 (12.1%))		(276 (22.2%))	
	FALSE	(188 (87.9%))		(965 (77.8%))	
NPI		0.34 (0.28–0.43)	186	0.41 (0.34–0.50)	1176
NPI_binary	TRUE	(170 (91.4%))		(1150 (97.7%))	
	FALSE	(16 (8.6%))		(27 (2.3%))	
NPIh		0.04 ((0.03–0.05)	185	0.05 (0.04–0.06)	1155
NI		1.24 (0.94–1.45)	214	1.87 (1.47–2.35)	1241
NI > 1.5	TRUE	(49 (22.9%))		(910 (73.3%))	
	FALSE	(165 (77.1%))		(331 (26.7%))	
NI > 1	TRUE	(146 (68.2%))		(1153 (92.9%))	
	FALSE	(68 (31.8%))		(88 (7.1%))	
Sleep Duration (h)		8.24 (7.52–8.98)	206	8.89 (8.15–9.67)	1247
Intake_Night (L)		0.00 (0.00–3.33)	85	0.00 (0.00–2.92)	803

FVC: Frequency volume chart; NUPh: nocturnal urine production per hour; NUPw: nocturnal urine production per weight; NPI: nocturnal polyuria index; NI: nocturia index; BMI: body mass index; # number.

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
