# Peer review of "Assessment of the Most Impactful Combination of Factors Associated with Nocturia and to Define Nocturnal Polyuria by Multivariate Modelling"

_jcm, 2020, doi:10.3390/jcm9072262_

Round 1

Reviewer 1 Report

This paper showed the statistical analyses to distinguish non-significant nocturia and significant nocturia by factors associated with “nocturnal polyuria” from data of three clinical trials. Although the comprehensive analyses using most of the definitions for nocturnal polyuria can be valuable to show their association strength with significant nocturia, some concerns I have, especially the clinical significance of the definition of nocturnal polyuria.

1) The nocturnal index (NI) was the most associated definition related with nocturia, but it was an expected result and it did not mean “nocturnal polyuria”, as the author mentioned “NI was not specific to excess of nightly urine production, as it also captured nocturia due to small FBC.” Since the nocturia is determined by the nocturnal frequency that is derived from nocturnal urine production and functional bladder capacities, it would be somewhat confusing to use the NI as the definition of nocturnal polyuria here. The authors recommended to use NUPh without applying threshold for disease as specific to an excess of nightly urine production for a general population that has a complaint of nocturia, for the understanding of each treatable contributor factor and in choice of effective treatment. However, it might not be appropriate in the clinical setting to consider the patient’s status from NUPh, even combined with bladder size and sleep duration, since global polyuria is important to think about the restriction of drinking water, which is quite often related to their lifestyle and is important factor to use desmopressin safely.  To think about nocturnal polyuria, daytime urine volume could not be ignored in the clinical setting. The participants of this study with an average BMI of 29, the range of urine volume 24HR was from 920.62 ml to 1875.83 ml, whose low amount of urine volume could not be representative for the general population who complain of nocturia, perhaps due to the candidate for clinical trials for desmopressin. I am somewhat afraid that the conclusion of this study might mislead the treatment of nocturia in this version.

2) It would be not clear to readers what factors were included in the multivariate models and what software was used for the analyses.

Reviewer 2 Report

The tables are so crowded and need to be separated from each other.

The authors did not explain the reasons behind excluding criteria.

The authors did no explain there intervention, was there any treatment offered

The relationship between BPH and nocturnia was not demonstrated well

Reviewer 3 Report

The authors evaluated to determine which combinations of factors that can accurately discriminate clinically significant nocturia in patients to facilitate clinical management and treatment decisions using multivariate analysis.

They concluded Nocturnal Urine Production per hour (NUPh) has previously been shown among elderly males to be essential in nocturia and a very valid measure of nocturnal polyuria. Large and independent dataset now confirm that it can be applied in an adult population with a complaint of nocturia covering both males and females.

Well written.

This is an interesting and impressed report. The contents are concise and easy to read.

Round 2

Reviewer 1 Report

I appreciate the correction and reply of the authors, and I agree to use continuous NUPh in the absence of 24-hour polyuria from the finding of this study.